# "We will need to build up the atmosphere of trust again": Service providers' perceptions of experiences of COVID-19 amongst resettled refugee adolescents

**Sarah R. Meyer**[1], **Ilana Seff**[2], **Alli Gillespie**[2], **Hannah Brumbaum**[2], **Najat Qushua**[2], **Lindsay Stark**[2]*

**1** The Institute for Medical Information Processing, Biometry, and Epidemiology, University of Munich, Munich, Germany, **2** The Brown School at Washington University in St. Louis, St. Louis, MO, United States of America

* lindsaystark@wustl.edu

**Data Availability Statement:** Based on discussion with our IRB, they reviewed our consent documents and have given permission for data to

## Abstract

Adolescent resettled refugees across the United States have been significantly impacted by the COVID-19 pandemic, through socio-economic stressors in households, disproportionate morbidity and mortality in immigrant communities, and social isolation and loss of learning due to school closures and the shift to online learning. The Study of Adolescent Lives after Migration to America [SALaMA] investigates the mental health and wellbeing of adolescents who come from–or who have parents who came from–the Middle East and North Africa [MENA] region and settled in the U.S. There is a gap in understanding of the experiences during the pandemic of MENA-background adolescents in the U.S. The objective of this study was to describe the perspective of educators and other school-affiliated service providers on the impact of the COVID-19 pandemic on mental health and wellbeing of adolescent resettled refugees and access to and quality of education and support services for adolescent resettled refugees. The researchers collected data using in-depth interviews with key informants in Chicago, Illinois; Harrisonburg, Virginia; and Detroit Metropolitan Area [DMA], Michigan, Key informants were school administrators, managers of English language learning services and programs, teachers, therapists, staff of non-governmental organizations and/ or community-based organizations, and case workers. Data analysis was conducted utilizing directed content analysis to develop an initial codebook and identify key themes in the data. Findings revealed a number of pathways through which the pandemic impacted adolescent refugees and immigrants' mental health and wellbeing, with online programming impacting students' engagement, motivation and social isolation in terms of peer and provider relationships. Specific dynamics in refugee adolescents' households increased stressors and reduced engagement through online learning, and access to space and resources needed to support learning during school closures were limited. Service providers emphasized multiple and overlapping impacts on service quality and access, resulting in reduced social supports and mental health prevention and response approaches. Due to the long-term impacts of school closures in the first two years of the

**Funding:** This study was funded by Qatar Foundation International, grant number G-7429377541, awarded to corresponding author LS. Qatar Foundation International's website may be found at: https://www.qfi.org/ The funders contributed to the study design, but did not contribute to data collection, analysis, publication decision or manuscript preparation.

**Competing interests:** The authors have declared that no competing interests exist.

pandemic, and ongoing disruption, these data both provide a snapshot of the impacts of the pandemic at a specific moment, as well as insights into ways forward in terms of adapting services and engaging students within restrictions and limitations due to the pandemic. These findings emphasize the need for educators and mental health service providers to rebuild and strengthen relationships with students and families. These findings indicate the need to consider, support and expand social support and mental health services, specifically for refugee adolescent students, in the context of learning and well-being during the COVID-19 pandemic.

## 1. Introduction

The COVID-19 pandemic continues to have an unprecedented global impact. While adolescents are less impacted clinically by the virus [1–3], this 10–19 year old age group [4] has been directly and indirectly affected by the pandemic through measures to control the virus, including quarantine policies, school closures and the shift to remote learning [5–7]. As lockdowns and school closures went into effect around the world, schools across all 50 U.S. states closed in March 2020 and millions of adolescents shifted to remote learning. As of December 2020, more than two thirds of U.S. students were still not attending school fully in person, contributing to learning loss across all grade levels [8]. These measures to curb the spread of the COVID-19 virus led to prolonged periods of social isolation, which can be particularly detrimental for adolescents due to the importance of peer groups and connectedness during this critical developmental period [9, 10]. Throughout the U.S., adolescents spent more time at home and had less access to external support services. Households experienced significant stress, often resulting in reduced family functioning. Physical and psychological maltreatment from household members and caregivers increased throughout the pandemic [5, 7, 11–15].

Adolescents in the U.S. have also been indirectly impacted by the pandemic in numerous ways. As of early 2022, the U.S. has reported in excess of 57 million cases of COVID-19 and over 827,000 associated deaths, with a cumulative confirmed case rate of 16,708 cases per 100,000 persons since the onset of the pandemic. [16, 17] As of April 30, 2021, more than 100,000 minors in the United States had lost at least one primary caregiver to COVID-19 [6]. In addition, many children and adolescents have described the impact of caregivers' stressors due to the pandemic, including financial insecurity, unemployment, health concerns, intimate partner violence, and substance use, on their wellbeing [18]. Mental health in adult and young adult populations has declined considerably with symptoms of depression and anxiety spiking in the first months of the pandemic; [19] increases in or new onset of substance use have also been observed [19] All of these pandemic-related phenomena have imposed new and evolving challenges upon adolescents, their education, their family and peer relationships, and their mental health.

Immigrants and refugees in the U.S. and other high-income countries have been particularly impacted by the COVID-19 pandemic, facing numerous and distinct risks that have compromised their health and welfare. Immigrant populations face especially high risks of infection by COVID-19, due largely to the compounding challenges of limited English proficiency, increased risk of exposure to COVID-19 due to employment in front-line services, and vulnerability to misinformation. [20–22] During the pandemic, immigrants and refugees in the U.S. often fell through the gaps in the social safety net, and did not receive many of the services intended to support vulnerable people in the U.S., including federal relief aid [23–25].

Parental unemployment, according to a 2021 study of five cities in the U.S., increased in migrant populations with 30% of immigrants included in the survey reporting losing a job during the pandemic [12]. These persistent and intensifying socioeconomic disadvantages, lower levels of health literacy, limited health care access, and higher risk of exposure to COVID-19 have made the pandemic a particularly precarious time for refugee and immigrant households.

These general dynamics of refugee and immigrant health during the COVID-19 pandemic have had distinct impacts for adolescents in refugee and immigrant households, particularly with respect to mental health. Adolescence is a life stage replete with risks and opportunities and when important physical, social and cognitive changes occur [26, 27]. Many mental health and psychosocial challenges have their onset during this developmental stage [28, 29]. During the COVID-19 pandemic, adolescents have seen rising rates of anxiety and less adequate sleep [30], and adolescents who were forced to isolate or quarantine experienced decreases in overall mental health and increased feelings of helplessness, worry, and fear [26]. Additionally, social isolation, which has increased due to quarantines and school closures, has been found to increase the risk of depression in adolescents [10]. This isolation and lack of social connection can lead to stress [31], and throughout the pandemic, rates of youth mental health conditions, including suicidal ideation, have increased [32]. Although research on COVID-19's impact on refugee mental health is limited, there is some evidence which suggests that COVID-19 has triggered traumatic memories in refugees, worsening their overall mental health [33], while some adolescent refugees, particularly those from the Middle East and North Africa (MENA) region already face higher risks of experiencing suicidal ideation associated with migration and stressful life events [34]. Alongside reinforcing existing health and social inequities, there is a risk that COVID-19 school closures disproportionately impact students who are already disadvantaged in terms of material support and personal and academic preparedness for home learning [24, 35, 36]. Disproportionate learning loss and educational attainment largely attributed to language barriers may contribute to the phenomenon of non-English speaking students in English-dominant communities being 50% more likely to be categorized as academically low-performing during the COVID-19 pandemic [24, 36–39].

Throughout implementation of the Study of Adolescent Lives after Migration to America [SALaMA] project, researchers have investigated the mental health and wellbeing of adolescents who come from–or who have parents who came from–the MENA region and settled in the U.S. Findings from the study indicate that alongside high levels of resilience and effective approaches to coping with challenges in resettlement contexts [40, 41], there are significantly increased levels of suicide ideation amongst adolescents born outside of the United States compared to U.S.-born students in quantitative studies in Detroit, Michigan and Harrisonburg, Virginia [34]. These students may be particularly vulnerable to the dynamics of COVID-19 and its impacts on economic stability, education quality and access, and mental health status. However, in the emerging evidence-base on the social, economic and mental health impacts of COVID-19 on students in the U.S., there is limited research focused specifically on refugee and immigrant youth, and nothing focused on youth from the MENA region. Some research has explored teachers' perspectives on the shift to remote learning, however, the specific challenges of supporting MENA-background students through the challenges of the pandemic have not been documented. Moreover, perspectives on the impact have primarily focused on the education sector and mental health system separately, rather than exploring how mental health supports usually in place through formal and informal practices in education were impacted by the pandemic.

The COVID-19 pandemic began during ongoing data collection for SALaMA. Interviews with teachers and other service providers for adolescent refugees in Chicago, Illinois,

Harrisonburg, Virginia and Detroit Metropolitan Area [DMA], Michigan, indicated the centrality of disruptions to education and services wrought by the COVID-19 pandemic, and the research team therefore incorporated additional questions focused on impacts of COVID-19 on resettled adolescent refugees into in-depth interviews conducted with educators and other service providers. The objective of this analysis, based on two phases of qualitative data collection for SALaMA, is to describe the perspective of educators and other school-affiliated service providers regarding experiences of the COVID-19 pandemic on mental health and wellbeing amongst adolescent resettled refugees and access to and quality of education and support services for adolescent resettled refugees.

## 2. Methods

Data were collected as part of the SALaMA study, a mixed methods study of mental health and psychosocial wellbeing of adolescent students from the MENA region, conducted in several cities across the U.S.

### Setting

The present analysis utilized data collected in DMA, Chicago, and Harrisonburg. These sites were selected purposively, based on considerations including the research team's existing relationships with school systems and the cities' histories of resettling refugees from the MENA region. Refugee resettlement data for the three states show that between October 2014 –September 2015, of 2,658 refugees resettled in Illinois, 811 were from MENA region. Among the 3,012 refugees resettled to Michigan in the same period, 1,653 were from the MENA region and of 1,312 refugees resettled to Virginia, 624 were from the MENA region [42]. In addition, individuals who received Special Immigrant Visas (individuals from Iraq or Afghanistan who worked with U.S. armed forced and are eligible for a specific resettlement visa) were resettled in these states.

### Sample

Key informants in all sites were involved in provision of education or mental health services for students, particularly students from the MENA region. Key informants included school administrators, managers of English language learning services and programs, teachers, therapists, staff of non-governmental organizations and/ or community-based organizations, and case workers. The sample was identified through a combination of purposive sampling and snowball sampling. In all sites, connections with specific schools and relevant organizations had been made in prior phases of SALaMA. In the first stage of research in Chicago, the research team conducted key informant interviews with school contacts as a first step, following which the school contacts provided names of services providers whom they knew of or worked with, both within and outside schools. In the second phase of interviews in Chicago, we revisited the list of key informants and conducted a follow-up interview with one key informant, while targeting more mental health service providers through key contacts in the community and local organizations. For the key informant interviews in Harrisonburg and DMA, the research team put together a list of relevant key informants who had participated in a previous phase of data collection in each site (2019 in DMA and 2018 in Harrisonburg), reached out to these key informants, and asked key study contacts in each site to add relevant individuals or update the list. Each potential respondent was reached via e mail. The research team sent a minimum of two follow-up emails to potential respondents who did not respond initially. Potential respondents sometimes indicated a colleague who would be better placed to participate in an interview on the study topics. Throughout data collection, each key informant was

**Table 1. Key informants by location, position and sex.**

|  | Chicago | Harrisonburg | Detroit Metropolitan Area | Total |
|---|---|---|---|---|
| Total Number of key informants | 25 | 11 | 10 | 46 |
| Male | 8 | 1 | 3 | 12 |
| Female | 17 | 10 | 7 | 34 |
| Affiliation |  |  |  |  |
| School | 11 | 3 | 2 | 16 |
| District | 3 | 7 | 0 | 10 |
| Community-based organization | 11 | 1 | 8 | 20 |
| Position |  |  |  |  |
| Service provider | 17 | 6 | 9 | 32 |
| Leadership | 8 | 5 | 1 | 14 |

asked if they could think of additional relevant key informants, and the research team reached out to these recommended individuals to request an interview. A total of 46 key informants were included in the two phases of data collection in Chicago, and in data collection in DMA and Harrisonburg (see Table 1).

## Data collection

Data collection took place in two separate phases, based on different in-depth interview guides. In the first phase of interviews in Chicago, key informants were interviewed between January and March 2021. The interview guide focused on understanding key informants' role in supporting MENA students in the classroom and school environment, types of services and supports available to MENA-background students, key informants' perceptions of the main barriers and facilitators of MENA-background students' wellbeing and learning, and perceptions of influences on MENA-background students, including home environments and peer relationships. Specific questions regarding the impact of COVID-19 on key informants' provision of services or their perceptions of impacts on students' wellbeing were not included. However, given the timing of the data collection, the impact of COVID-19 emerged as a key theme and thus these data are included in the current analysis. For data collection in Harrisonburg and Detroit, and a second phase of data collection in Chicago, the interview guide focused on findings regarding MENA students' elevated risk of suicide ideation, which had been identified in a phase of quantitative research conducted as part of SALaMA [34]. This interview guide included specific questions regarding how COVID-19 had impacted students' wellbeing, availability and provision of services and supports, and students' help-seeking behaviors. Interviews in the DMA and Harrisonburg were conducted from March to September 2021 and the second phase of interviews in Chicago were conducted from August to October 2021.

All interviews were conducted via Zoom, with either one or two trained qualitative interviewers conducting the interview. Respondents were sent a consent form via the Qualtrics online platform, and read and completed the written consent form prior to the interview. All interviews, apart from one, were audio-recorded and then transcribed. One respondent chose not to be audio-recorded, and for that interview, a second interviewer took detailed notes and a transcript was created from the notes following the interview. The majority of interviews included in this analysis were conducted during the period of full-time virtual learning in schools, however, some interviews in the second phase of Chicago interviews were conducted when students had returned to in-person learning.

## Data analyses

Data analysis was conducted in several phases. Based on the objectives of the study, the research team employed directed content analysis. Directed content analysis was selected as the analytic method as it fit the objectives of the study; one of the benefits of this approach is that it can be used to expand on or further describe a phenomenon or set of phenomena about which there is already some theory or research [43]. In the case of this study, there was some prior research conducted as part of the SALaMA project that indicated the relevance of these research questions. While directed content analysis may result in bias due to use of a predetermined theory or categories, in the case of the present analysis, the initial codes were drawn directly from data.

Using this approach, in the first phase, the research team reviewed all transcripts and identified preliminary themes [43]. Based on these themes, the team developed an initial codebook. To test and refine the codebook, the team employed deductive coding, which entailed selecting a single interview for all research team members to code, and in addition, pairs of research team members coded an additional interview, using Dedoose qualitative analysis software. Through this process, the team identified similarities and differences in utilization of the codebook and combined or added codes as needed. Following further revisions to the codebook, the research team re-coded the same interview, reviewed any additional discrepancies, and finalized the codebook based on further discussions. Following this, the research team–which included individuals with training in qualitative research methods, public health, social work and international development–applied the codebook to the entire dataset. At this stage, inductive coding was utilized, using the raw data to inform new or revised codes. The research team drew on Nowell et al.'s guidance regarding establishing trustworthiness through the process of thematic analysis in several ways [44]. For example, at the phase of data familiarization, the research team read and re-read interview transcripts, taking notes on potential codes, themes and interconnections. In the phase of generating initial codes, the research team kept a careful audit trail of all potential codes and codes that had been combined or deleted. In the phase of searching for themes, the team met and discussed how particular codes combined into larger themes, and all data analysis team members reviewed and approved the final coding and thematic scheme.

Based on emerging themes, frequency of code application and relevance to the key research questions, the following codes were exported from Dedoose for excerpt analysis: basic needs, caregiver engagement and supervision, online programming, student mental health, student engagement, family wellbeing, social isolation, decline in quality of programs; modes of delivery and outreach; innovative approaches; and psychosocial outreach (see Table 2 for primary thematic codes and sub-codes).

## Ethical approvals

The research protocol was approved by the Institutional Review Board at Washington University in St. Louis (IRB ID# 201902044). All audio and written transcripts were assigned an identification number, which was only linked to respondents' names in a password-protected file.

## Results

### Service providers' perspectives on the mental health and wellbeing of adolescent refugees and immigrants during COVID-19 pandemic

Data revealed a number of pathways through which the pandemic impacted adolescent refugees and immigrants' mental health and wellbeing. Key informants highlighted how online

Table 2.  Primary codes and sub-codes.

| Parent Code | Child Codes | Grandchild Codes | Definition | Examples |
|---|---|---|---|---|
| | | | Use when participants describe the process or experience of navigating online programming. *ALWAYS co-code with who is impacted by online programming (eg: Impact on Students, Impact on Teachers, Impact on Families) Co-code with "Basic Needs" if equipment and WiFi are mentioned. Co-code with "Zoom fatigue" if the specific negative experiences of students with Zoom and other on-line platforms is described | "And it's harder now to reach them you know, before I could just like if if anybody saw them coming into the building I can just put out an alert and boom the second he walks in the building he's stopped at the front they call me down to go see him. But now, he doesn't even have to show up to class if he doesn't want to you know like [laughs] he can just ghost it and I guess he'll probably pop up in two weeks and tell us oh you know this this this and that. Can I please have my my uh passing grades. Oy." (C1.002) "The challenge too, is, you know, it's really hard to deliver some services that you would normally deliver because you're in a remote environment. So we also have to be conscious of screen time for example. Like we want, we might want to run a lot of tutoring, but at the same time we've got to realize these kids have already been hours and hours on a computer in front of a screen, and so more time that day is probably not necessarily beneficial." (C1.015) |
| Impact on families/ households | Family well-being | | When participants describe the impact of COVID-19 pandemic and associated control measures on overall well-being of families | "They don't like to talk. They don't like to see friends outside. Afraid. Also, their families—they are scared about COVID. Um. . .sometimes if they want to visit their homes, they are more scared because they will be so close and in the beginning, they don't like them to go out so they are in a trap. Kind of trap." (HS1.001) |
| | Basic Needs | | Any reference to basic or survival needs being impacted during the pandemic, including: housing and home environment, lack of equipment and poor or no Wifi access, food access and security. | "Um, a lot of families you know, who were doing well before ended up reverting back to some survival needs. So it became getting folks food, and in the summer getting kids set up with tech so that they could log in for school, understanding some works, right, all those things" (C1.012) "I think also the biggest thing that I'm concerned about for families is like rent and paying rent and really eviction moratorium and when that ends, what that's gonna look like for families." (C1.004) "And if you have, if you have a good home environment, maybe it's okay, but if your environment home is like really difficult then you're just stuck." (MS1.001) |
| | Caregiver engagement and supervision | | Any reference to caregivers' preference for or ability to engage and supervise their children at home during the pandemic. Can include discussions of overseeing the use of technology, the need to work outside of the home, how involved caregivers are in children's schooling and social lives, etc. | "I mean, and these kids, I'm thinking, ranged in age from maybe. . .first to. . .maybe fifth grade? Um, and their parent, you know th-the, mother was going to be home soon. How long had they been home by themselves? I don't know. Um, my sense was, is that they were definitely instructed not to open the door, and decided that I looked safe and so they did. Um, and then when I came back later, um, in the evening, um, I said 'that's okay, I'll just come back', you know. Um, and when I came back around 8:00, um, and t-they were outside playing, and, you know, the parents were there but it's just. . .you know, lots of kids, home alone." (HS1.002) "So, it's very different, and of course staying at home can result in to over thinking can result into negative thoughts and so this might lead into suicidal thoughts um and what it really depends on the family situation fam family dynamics and if parents are socializing with their kids or not yeah." (MS1.002) |

*(Continued)*

**Table 2.** (Continued)

| Parent Code | Child Codes | | Grandchild Codes | Definition | Examples |
|---|---|---|---|---|---|
| **Service access and quality** | Academics | Student engagement and attendance | Zoom fatigue | *Student attendance and engagement*: Use when service providers comment on students' attendance and engagement with academics during the pandemic. This might include whether or not students log in to virtual classes, if they have their cameras on or off, if they participate in discussions or activities, and whether or not they are reachable by school staff. Include discussions of students falling off the radar, not achieving at the levels that they had prior to the pandemic.<br>*Zoom fatigue*: Use when service providers comment on students' lack of engagement, specifically with Zoom (or other online platforms used for education during the pandemic). | "Uh, I think that we're, we're very concerned about students who have seemed to have gone silent in this COVID period and with, with whom we've had very limited contact in any form. Maybe they've checked out of academic work. . . .um, often have checked out of other types of meaningful contact with, with staff. And so it's hard to know what they need and if they're being served in the way they deserve and if, if they're well, I mean at a basic level." (HS1.004)<br>"English language learner girls currently have the lowest, is the population at the school that has the lowest attendance and engagement with the school. I don't know why that is, but they are showing up the least apparently at Sullivan. Like I said, I just heard this the other day I don't quite know how that plays out statistically but um they, I think also the biggest thing that I'm concerned about for families is like rent and paying rent and really eviction moratorium and when that ends, what that's gonna look like for families." (C1.006) |
| | Reduction/closure of extra-curricular activities & programming | | | When participants describe a reduction or closure of extracurricular activities and programming as a result of the pandemic, including student groups, clubs, organization-led after school programs, sports, and the arts. | "Yeah, so we had one uh, we called it our speakeasy school, um, because uh, yeah none of our kids were you know, were able to go to school so we would pick up a family at a time and bring them to our after-school program area which is the basement of the church. Um, <laugh> and we'd just have secret school and we would um, help our kids with any of the technology hiccups or make sure that they're sticking to the schedule." (C1.014)<br>"Well, I mean there's obviously like less opportunities. There is, like less, you know programs, more disconnection like from things that were supportive. Um, so I'm like very concerned. I mean I didn't even do my program that I was going to do in the spring, because of COVID, right?" (MS1.001) |
| | Decrease in quality of programs and services | | | Use when participants describe how the administration and implementation of programs and services decreased in quality as a result of the pandemic. This could include limited equipment, technological difficulties, equity of provision, staffing shortages, unclear streams of communication, etc. | "And so turn over there has been, we in the past year during COVID have not been able to keep that, like, those relationships going. So like, I don't even know for example, right now who I would even contact at Senn, right? Like I don't have anyone's ear there right now. And that's probably on me in many ways, but um, they've also had a lot of admin turnover in the past 5 years too but um." (C1.011) |
| | Psychosocial support | | | Use when participants describe specific approaches to meeting the psychosocial needs of families, students and broader community | "Um, how can be maintained. I mean, I think you have to really actively try to create, um these like opportunities. So, what I always what I did in my class, um which is a college level class um so it a bit different different you know group of students, but um uh like we would always ever always make time for like a check in at the beginning and we put them I put them into breakout rooms. Um, so it's like smaller groups. So, they really did get to know each other more, and they would you know just kind of be able to like talk about sometimes kind of class related things but, like you know, a little more of like accessible it's not like technical and challenging." (MS1.001) |

(*Continued*)

**Table 2.** (Continued)

| Parent Code | Child Codes | Grandchild Codes | Definition | Examples |
|---|---|---|---|---|
| | Modes of outreach and service delivery | | Use when participants describe new modes of outreach and service delivery during the pandemic that aren't technology related, such as home visits, speakeasy school, etc. | "I was, like parents, I was calling parents and saying like, 'hi how can I help you'. They're saying I can't get my child, I can't change, I need to change my child's password I don't know how. So, I literally had to screen shot and help the dad and I was like I don't speak Arabic so I can't really do too much, um, in order to really like, find, like to really navigate. So I'm like let me see what I can do so I'm like this is what you're going to do first. So, I sent them screenshots to walk them through. So then that facilitated a conversation with IT. Like, yes, we have it in English and Spanish but what other languages do we have these tech, um, guides in? And then that prompted = Name = to say hey let's get them translated in our top languages, so we worked as a team to kind of move that faster um." (C1.007) |
| **Impact on students** | Academic Impact | Language learning / being a non-native learner during COVID / loss of English language | *Academic impact*: When participants describe how COVID-19 impacted students as it related to their academic motivation, performance, or development.<br>*Language learning / being a non-native learner during COVID / loss of English language*: When participants describe how COVID-19 impacted English language learning for newcomer students. | *Academic impact*: "Yes. . .often. I think that—uh, some of my colleagues have reflected that it seems like everybody kind of went down a step from where they would normally be this year. Or many, or most people, then—I shouldn't say everybody. So your 'A' student might be a 'B' student. and your 'C' student might be a 'D' student. And your student who is just hanging on now just fell off the cliff." (HS1.004)<br>*Language learning / being a non-native learner during COVID / loss of language*: "They are eager to learn English, that I know. And um schools, I mean, clearly there's a place to learn English, um, but what I'm recognizing and one of the challenges we're facing right now in this pandemic is around the English language. What the students don't have the opportunity necessarily for is the conversation in the hallway, the conversation in the lunchroom, I ran into a friend in the bathroom. We're using my English, we're practicing English, there's an afterschool sport that I like to participate in. there's all these different opportunities for kids to learn the language and those opportunities are so limited. They have their classes, they have their teacher, they have their lessons and all of those, but it's these additional opportunities that just increase the amount of practice the kids get with the language that they just are necessarily have access to. So that's one thing that I'm really worried about right now." (C1.005) |
| | Student mental health challenges during the pandemic | | Use when participants describe students' mental health challenges during the pandemic, including mentions of depression, anxiety, trauma, or specific symptoms associated with mental health. | "So that's when we start to see that sort of splinter into these more feelings of doom. Like honestly that's how, just felt unending, right. And I think that girls in the beginning, there was a lot of optimism and by the end of summer, that was gone. So it felt, we've seen the fatigue is one thing. But like, they're just, they have no optimism left. Like, they're just like, is this going to be forever?" (C1.011) |

(*Continued*)

**Table 2.** (Continued)

| Parent Code | Child Codes | Grandchild Codes | Definition | Examples |
|---|---|---|---|---|
| | Social isolation | | Use any time participants mention students feeling isolated, lonely, or less social, during the pandemic. This can be in relation to peers, family members, community members, or school staff. | "Yeah, as I told you—isolated and feel lonely. One of them called me, she said Mrs. R_, I feel I am alone. Although I am in the middle of the campus, and I have many students I can meet. . . but I feel that I am so isolated. Especially I can't talk to them and show them that I am not, like, zombie. And we can talk, we can share. . . so she said "I spent all the time in my room or the classroom sometimes when it is not on zoom."" (HS1.001) |
| Reactions / Strategies to respond to COVID | Innovative support strategies or approaches | | Ways in which providers facilitated ongoing educational needs or supports during the pandemic, including adjusting pedagogical approaches, offering new programming or services, or implementing new strategies to support students in their learning and wellbeing. | "I mean, I think there's been a huge effort by, um, by, um, our school system to make sure that there's virtual access, um, and that was just, that was just a lot of work just to make that happen. Um, and, um, and the home school liaisons have been, um, strategic this year in terms of really, making sure that that happens and finding out from families if they had access. And, and, you know, checking back in and making sure they have [mega] spots, or they had, you know, minimal-whatever they needed t-to make that happen." (HS1.002) "You know, we there was one um one time this was really cool some of our counselors got together with the it was a outside thing prior to re-opening. Um, they got together and they opened up an art um and are session group for students to paint backdrops, because those students didn't have it have an option so they didn't have a private space. So, you could kind of creative backdrop that you know so people couldn't see into their bedroom or wherever you know because that that was challenging too." (HS1.006) "So we've been trying to do more recorded psychoeducation videos, um, trying to check in, in ways that don't look like one hour of therapy sessions every week. You know we have monthly check-in lists or biweekly check-in lists or all of that looks different, but it's definitely affected kids." (C1.012) |

programming impacted students' engagement, motivation and social isolation in terms of peer and provider relationships—impacts which were particularly heightened for recently arrived students.

**Online programming.** Key informants perceived an increase in student mental health challenges for immigrant and refugee students as a result of virtual learning, as well as an impact on students' psychosocial wellbeing in terms of their overall engagement and motivation. A school principal in Michigan stated that the switch from in-person to online learning "*influenced the level of mental health problems*" they saw emerging in students. In response, this school hosted several virtual workshops about mental health. These workshops emphasized "*mental health awareness*" messages such as "*it is okay to have those thoughts, [. . .] it is okay to be depressed, it is okay to feel down*" (MS1.008). Another key informant in Harrisonburg who provided support services for refugee families shared how one parent was "*really concerned about her daughter*" and hoped the school could provide "*some in-person instruction*" as the daughter "*had not at all been successful in virtual learning [. . .], dealt with depression and had also been hospitalized*" (HS1.002).

Across sites, key informants lamented students' decreased motivation to attend classes in a virtual format, and discussed ways in which this impacted immigrant and refugee students in particular. Respondents felt that students' lack of engagement in virtual learning had a strong impact on their psychosocial wellbeing. A Harrisonburg-based home school liaison for refugee families noted that "*The use of zoom caused a decline in their academic performance as well as caused the deterioration of students' self-confidence*" (HS1.010).

Service providers described varying levels of student engagement throughout online programming, though, as one Harrisonburg student support provider indicated, they were most concerned with the many students who "*didn't even have their camera on or did not engage at all*," as it was "*just so hard to get a pulse on how* [these] *students were doing*" (HS1.007). A Chicago provider also spoke to the range of student engagement:

> *I'm thinking of a couple of them, who come to class every time, but they never have their camera on, they never talk,* [. . .] *a couple of them never turn in work, some of the other ones do, they don't respond to—when I address them directly or to emails, or to private messages and we can't get ahold of anybody at home. . . then other students that you know are very shy— and it's so easy online to hide behind the camera* (C1.017).

Another Chicago provider provided the example of an Arabic-speaking student who "*just says he doesn't know how to do the programs, and I keep telling him look, I'll work with you, I'll walk you through it, and he just, his family had told me 'no, he's smart, he can do it, he just doesn't want to'*" (C1.018). Providers emphasized how more recent arrivals faced added challenges compared to those who had been in the U.S. for longer. For example, a school-based provider in Michigan indicated that newer incoming students who did not speak much English yet experienced challenges *"logging into the computer [. . .] because they didn't know how"* (MS1.004). Similarly, a respondent in leadership at a Chicago school pointed out how *"some of them are brand new to the country and they never even used a computer before. So, you're expecting them to just jumpstart 'hi, come to school, pick up a computer and get started'"* (C1.007).

**Social isolation.**   Service providers identified social isolation—both from peers and trusted adults outside of students' families—as a major threat to refugee students' mental health and psychosocial wellbeing. Service providers explained the loneliness these adolescents felt in being separated from their peers, which at times led to symptoms of mental distress. A provider in Michigan described how caregivers reached out to them because their child *"won't even come out of their room*," and experienced worsened depression *"clearly because of COVID*" (MS1.004). While students of all backgrounds experienced social isolation, providers believed that the impacts were compounded for newcomer students. A therapist at an organization providing mental health services for refugees in Chicago explained that this exacerbated impact was because "*(A) they were maybe already a bit more isolated to begin with potentially; (B) they have less access to technology. . .technology and the, you know, that allows them to stay connected*" (CS1.006). Another Chicago-based refugee services manager noted, that for those who "*came sort of right as COVID started—February, March of 2020* [. . .] *their only experience of the United States has been their apartment* [. . .] *through the lens of COVID and quarantine*," which meant that "*learning English*" and "*making social connections has been a lot more challenging*" (CS1.001).

Providers highlighted how virtual connections created in response to social isolation from peers were often far less meaningful. A Chicago school-based service provider explained that students would need to "*start from scratch again*" to help them "*establish relationships*" and "*build up the atmosphere of trust again*" because "*there is not much personal touch*" to

relationships happening via "*the computer* screen" (C1.004). Providers explained that one factor contributing to exacerbated mental health challenges as a result of social isolation was that students were unaware that others were going through similar experiences. The Chicago-based refugee services manager explained that they "*had a couple kids say- ask me, like, 'do other people feel this way? Do other kids feel this way? Do other people my age feel this way?'*" (CS1.001). The respondent continued:

> *And when you're in school, you can potentially connect with people who do, right? You don't have to ask, other people will tell you, right? Your peers will tell you that they're also feeling that way. So that compounds, right? And you feel even more isolated, because you feel like it's a singular experience.*

On the other hand, a provider in Michigan explained that some students assumed everyone was going through the same mental health challenges that they were, which "*causes you not to reach out for assistance because* [. . .] *just being helpless and thinking that it's okay to be like that caused a lot of kids to probably not seek out for help or not even care about it, either*" (MS1.005).

Providers perceived a decrease in help-seeking behaviors to be connected to students' increased social isolation, not just from their peers, but from trusted adults outside of their family, including adults in their cultural community, teachers, and other school-based support staff. A provider in Chicago emphasized that families often relied on "*a village of adults to be able to support [MENA students]*" (CS1.001) and in Harrisonburg, a provider described their attempts at "*sustained and repeated outreach*" and felt that "*some students clearly chose to ignore the outreach for whatever reason it was,*" explaining that there were some "*hardcore want-to-disappear-off-the-face-of-the-earth kids*" (HS1.004).

**Family wellbeing and access to services.**   Respondents indicated that students' wellbeing was largely influenced by the collective wellbeing within their households during the pandemic. Household-level stressors influenced students' engagement in remote learning and other virtual services. Lack of space and lack of internet connectivity contributed to students' and caregivers' difficulties; however, providers also identified some benefits of virtual programming.

**Household roles and stressors.**   Key informants described many barriers to caregivers' participation and engagement with computer-based education. These perceived barriers included some caregivers' employment obligations that meant students were home alone during the school day, school information being provided only in English or Spanish and not in caregivers' own language, and challenges navigating technology. Providers explained how parents were forced to take on additional roles to support their children in schooling and provision of support that peers or teachers might usually provide. A principal of a high school in Michigan noted factors that "*contributed to the stress of the student,*" and emphasized:

> *I mean, parents all of a sudden, all of a sudden, they were expected to be their child's teacher. Somebody who might not even speak the language! Somebody who does not know the school system here. So umm, parents did not- were not trained to be teachers! They were trained to be parents* (MS1.008)

Another informant in Chicago acknowledged these added burdens by questioning, "*Parents are not doctors. So why are they all of a sudden therapists?*" (CS1.004).

Conversely, providers explained that many students had added responsibilities within the home due to some caregivers' expectation that their children should support household chores or childcare at the same time as engaging in virtual schooling. For example, a student in

Michigan "*was having a very hard time doing her schoolwork online*" and was unable to make up the work in summer school "*because her mom had just had a baby, and she had to stay home and take care of the younger siblings. Um, and, you know, help out with mom,*" according to one key informant (MS1.004). An educator in Chicago explained that "*some of the kids are really stressed out because they are working a double duty*" (C1.018). They elaborated on a phone call they had with a parent "*of one of my Arabic students:*":

> *The kid had been not turning in assignments as much and when he did, they were really not that good [. . .] you know, he's logged on but he's not there. [. . .] I asked [. . .] does he help around the house when he's supposed to be working in my class? [. . .] I call his name and a lot of times he says, "oh I'm sorry I was in the basement I had to get the laundry. Or I was helping my mom with the cooking." [. . .] And she said "yeah, I was sick, so he was helping me." And so, I don't think that the parents sometimes realize, "hey this is really school," you know, "this is very important that you do this."* (C1.018)

Key informants perceived a bi-directional relationship between student mental health and wellbeing and family stability and wellbeing. In other words, students' poor mental health contributed to poor family dynamics and at the same time, home stressors led to students' decreased psychosocial wellbeing. Respondents noted that while many of these dynamics existed in refugee and immigrant students' households prior to the pandemic, the pandemic had exacerbated the stressors. A Harrisonburg-based home school liaison felt that "*the gap between the students and their parents has become larger because they have been isolated for a long time in their rooms*" (HS1.010). A provider in Chicago noted escalating tension within homes and families "*coming to us much more often in crisis, whereas before we might have been able to see something escalating [. . .] because someone saw it at school*" (CS1.001).

Tensions within households were sometimes related to family challenges acquiring or meeting basic needs during the pandemic. One key informant in Michigan shared that they filled the gap of students who "*did not have food at home*" by sending "*our buses with food to the homes*" (MS1.008). When newcomer students did seek help from providers, it was often around basic needs. A service provider at a refugee-focused non-governmental organization in Chicago shared that a "*big challenge*" was supporting caregivers who were unemployed during the pandemic; the provider "*had a lot of youth calling me and saying, 'can you help me apply for my parent's unemployment?'*" (C1.012).

**Family wellbeing and access to learning.** Many respondents noted that refugee adolescents faced challenges in terms of access to remote learning due to lack of internet connectivity, limited devices available for remote learning, and lack of space for remote learning in their households. Access issues were largely compounded by household socio-economic status and family wellbeing. One respondent noted challenges for refugee students to find a private space for therapy sessions; working with one Syrian student, the key informant shared that they had "*to find a space that [the student] can go and have [their] private space. . .so she sits in her bathroom and that's where she can have privacy*" (C1.012). For other respondents, the concern that some students might be embarrassed to turn on their cameras during online learning due to their household environment emerged. One respondent explained,

> "*we don't know the situation at home so sometimes we're forcing them to turn on their cameras um and they don't want. You know couple of times I saw, you know, what's going on, how many people live in the household and um what do they have like furniture-wise or whatever. So, the level of noise at home is unbelievable, that's why they always majority of them are mute*" (C1.004)

Refugee adolescents were perceived as more likely to experience structural barriers to remote learning and other virtual modalities of services. Many refugee students needed additional support to navigate different online platforms, passwords and how to sign into links, which was "*very different for the rest of the population because they're very familiar with that technology*" (C1.008). The demands of remote learning were perceived as heightened for English language learner students, as "*manag[ing] five tabs at a given time*" is difficult (C1.010).

At the same time, according to some respondents, modes of service delivery necessitated by the pandemic were preferable for some refugee students. One Michigan-based case worker at a community-based organization explained that for some of these adolescents, the school environment was very stressful–"*I have to leave my house, I have to go into place after where people talk a different language, different food. . . you have people that might make fun of the way I dress or where talk*" (MS1.003). For such students, remote learning helped reduce anxiety associated with attending school, while enabling students to remain engaged in their education.

## Impacts on service quality in schools

Providers noted the multiple and overlapping ways in which the COVID-19 pandemic impacted service quality, including capacity to provide specific mental health and language support programs and reduced quality of online programming compared to in-person services. Educators and service providers emphasized that the usual services provided to refugee adolescents, ranging from specific language learning supports to individualized outreach for students facing mental health challenges, were compromised by school closures and reductions of in-person programming. Extra-curricular activities, specific events to celebrate and support refugee students, and classroom-based innovations, such as mindfulness sessions with students, were canceled. One educator in Harrisonburg explained that in terms of extra-curricular provisions, there was "*not much being done,*" noting that the major efforts were primarily focused on retaining students in online learning, "*just trying to keep kids in school, and make sure that kids have access. . .that was the effort this year, right?*" (HS1.002). Referring to an after-school tutoring program, one respondent noted, "*it was a big loss, I'd say, during 2020. . .to not have that*" (C1.014). Respondents routinely expressed disappointment and frustration at not being able to provide the range of services, or the level of service quality, that they had previously been able to provide prior to COVID-19.

Respondents described a context in which the focus of their work was on ensuring an effective shift to online modalities, including provision of therapy and regular classroom instruction. The additional supports and outreach that they usually provided for refugee adolescents were significantly impacted by the pressure to keep programming afloat amidst the upheavals of suspended in-person programming. Respondents expressed concern about the impact of this reduction of support, including that students' essential needs were not being met:

> "*it's just during that call to a family or to a kid who you haven't heard from in a while, it's like ehh what's going on. . .I think the nightmare is that you neglect someone and then they fall off the deep end*" (C1.014).

Respondents narrated multiple and intense efforts to reach students who they were concerned about during the school closures. Educators and other service providers explained that online contact with students and their families impacted the quality of their interactions, for example, one respondent described "*those things. . .left unsaid*" in an online conversation, explaining that in-person,

*"sometimes you're more patient about asking all the questions and making sure you've covered everything, but when you're on a video it feels uncomfortable so sometimes you might forget, you know, family might forget to ask something, or make you aware of something"* (C1.015).

Respondents expressed concerns that the specific needs of refugee adolescent students were more likely to be missed during the pandemic, with limitations on in-person interactions and reduced knowledge of individual students' situations.

In particular, providers of mental health services noted that the usual referral networks between schools and mental health supports for students were severely affected by school closures. For example, a home school liaison in Harrisonburg explained how they were usually made aware of student needs "*when there is a problem inside the school, for example when they fight with their friends.*" The provider described a "*culture of talking to friends or neighbors when [students are] upset*" and not seeking "*help from mental health providers.*" This contributed to fewer referrals for "*sadness or psychological distress*" during the pandemic as students could not seek immediate, in-person support (HS1.010).

The quality of mental health services provided by telephone or through online platforms was widely perceived as reduced; one respondent indicated that students were "*less likely to seek therapy. . .via zoom*" (MS1.009). A key informant in Harrisonburg whose role focused on teaching and learning noted that efforts were made to hold sessions focusing on mental health, explaining that there was,

*"mental health information and some space for talking, but not a—you know, on a screen with 20 squares, many of them dark. . .it doesn't feel like a really safe place to be expressive of difficult topics"* (HS1.004).

A Chicago-based mental health service provider similarly noticed that "*the kids are not reaching out to us*" for online support because they knew they would receive "*a very disconnected format of talking or connecting.*" They described this lack of connection as "*terrible*" compared to when students were able to "*literally* [. . .] *walk down the hallway to someone that they trusted*" to ask for help (CS1.002). One Chicago-based service provider stated,

*"there's no replacement for being in the classroom and having that one-on-one support that you get with your teacher and, having that accountability for doing your homework and participating and showing up on time. . . the lack of structure has been quite damaging"* (C1.013).

One respondent blamed the "*stupid screen*" (C1.003) as building a barrier between teachers and students, both in terms of student participation in learning and educators' ability to form and maintain strong personal relationships with students.

## Innovations in service delivery and student support

Service providers described a number of innovative approaches to support and engagement undertaken during the pandemic, as well as changes to modalities of service delivery and approaches to engaging students that they hoped to retain in the future.

### Innovative approaches towards support and engagement

In the context of massive disruption to in-person programming, including classroom teaching, afterschool programming, and targeted mental health supports, service providers described a

range of different innovations in service delivery aimed to increase or maintain access and quality, and ensure student and family engagement. Respondents described psychoeducation videos, holding parents' and students' meetings outside (when school closures were in effect but COVID numbers were low enough to enable outdoor interactions), online programming on maintaining hope during the pandemic, and teacher support programming. Respondents emphasized creative and intensive efforts to orient students and families to remote learning; as one respondent explained, *"I literally was teaching girls how to get into Zoom on sidewalks all summer"* (C1.011). One respondent explained that the pandemic had brought teacher wellbeing to the forefront in a more meaningful way, explaining that before COVID-19, *"wellness was much, much more siloed and much more superficial, you know, like someone would ask me– Hey, what do you do to decompress?. . .since the pandemic. . ..there's been a big spotlight on teacher wellness"* (C1.003). The school leader continued, stating that teacher wellbeing was essential to ensuring student wellbeing as *"if you don't feed the teachers, they'll eat the kids"* (C1.003). Educators perceived teacher support programs as vital to students' wellbeing and learning, given that the enormous pressures on teachers during the pandemic were compromising quality of teaching. Similarly, some respondents noted that the pandemic had spotlighted the importance of student mental health, coping mechanisms and approaches to reducing stress, and embedded within overall school approaches efforts to support students in a more holistic way than was available prior to the pandemic. One respondent explained that open sessions for students to discuss stressors and challenges had been a vital "*outlet to talk*" *during the pandemic*. She noted,

> *"Whether it was the [George] Floyd protests, or the pandemic, you know. . . [it's important to] just having some sort of programming that is very unintrusive and, and you know, very flexible and open. And allowing the kids to breathe and vent"* (C1.014).

**Approach and work with refugee students going forward.**   While the majority of respondents noted the widening gaps in academic achievement between refugee students and other students due to COVID-19, a minority of respondents noted that the ways in which refugee students had coped, and in some cases, thrived, indicated a need to shift their approach to these students. One respondent explained that she had found that *"some of our students have even academically thrived in the absence of some of the other pressures of being an adolescent in school,"* and that for refugee students specifically, the pandemic and changes in service delivery and modalities of access indicated that

> *"sometimes we've scaffolded so much of their experience, that we never remove the scaffolds. . . we've seen this year that many of our students are capable of things that, uh, you know, without having an adult hovering by their shoulder. . .I hope that we will remember that and can, and can continue to support their independence as learners as much as we can"* (HS1.004).

For another respondent, one of the key results of the pandemic was an understanding that *"it just can't be one size fits all,"* and that for some students, their experience of learning or therapy behind a screen or on the phone might be better than that of in-person (C1.004).

## Discussion

These qualitative data provide insight into the experiences of service providers offering education and mental health services to refugee and immigrant students during the first eighteen months of the COVID-19 pandemic in the U.S. Data for this study were collected at a time

when return to in-person schooling seemed imminent. Participants who were expecting to return to 'normal' reflected on the impacts of COVID-19, thus they include perspectives of how experiences providing services during the pandemic could be brought forward towards improving teaching and engagement with students from the MENA region in the future. The findings emphasize the need for educators and mental health service providers to rebuild and strengthen relationships with students and families.

This analysis also explores service providers' perceptions of the association between online programming and mental health challenges for refugee and immigrant students. Multiple contradictions emerged regarding online programming, participation and wellbeing; for some service providers, the pandemic spurred much-needed independence and self-sufficiency for MENA-background students, while for others, students' nearly total lack of participation in online programming led to immense learning loss and social isolation. Nearly three years into the pandemic, there remains the dual challenge of protecting public health through measures to control the spread of the COVID-19 virus, while simultaneously supporting and promoting students' mental health. The 2021 U.S. Surgeon General's Advisory, *Protecting Youth Mental Health*, addressed the impact of the pandemic on youth mental health, stating that the "unfathomable number of deaths, pervasive sense of fear, economic instability, and forced physical distancing from loved ones, friends, and communities have exacerbated the unprecedented stresses young people already faced" [45]. Previous research conducted as part of SALaMA indicates that there are significantly increased levels of suicide ideation amongst adolescents born outside of the United States compared to U.S.-born students in Detroit, Michigan and Harrisonburg, Virginia [34]. Student help-seeking behaviors, integrated supports and mental health promotion provided in the classroom, and high-quality, targeted formal health services were significantly impacted by the pandemic. Critical factors influencing psychosocial health outcomes among this population–including school acceptance and belonging, language support and peer relationships–have been impacted and eroded during the pandemic. These findings highlight the need to ensure that acute needs that may have emerged during the pandemic are identified and addressed in the course of education and mental health service provision, whether in-person or online.

The refugee experience, intertwined with socio-economic status, compounded the negative impacts of COVID-19 for individuals and households. Prior research conducted as part of SALaMA and existing data show that prior to COVID-19, MENA-background students faced a range of stressors at school and at home and did not enter the pandemic with the same social, economic and cultural resources as other students. In this study, key informants reported that they recognized that MENA-background students were, prior to the pandemic, faced with significant challenges in integration into learning contexts. The pandemic brought to light how these challenges intersect with other inequities, and service providers reported that refugee adolescents were less able to engage in online learning and more likely to experience significant stressors at home, including challenges meeting basic needs. While the compounded impacts of the pandemic on the health and socioeconomic well-being of immigrants and refugees in the U.S. have been highlighted previously [46], the present data reveal the extent to which service providers struggled to identify and employ approaches to adequately address these challenges for their students. These findings are also echoed in research on adolescent resettled refugees' experiences in Canada and Australia during the pandemic [36, 47], as well as other resettled refugee populations in the U.S, such as Rohingya refugees [48].

Evidence indicates that student and caregivers' engagement in the learning process and school environment is key to achieving positive developmental and desired academic outcomes [49–51]. Relationship-building–with teachers, peers and other trusted adults within and beyond the school system–is key for student engagement and wellbeing. The closure of

schools and reduction in youth programs during the COVID-19 pandemic intensified challenges faced by service providers in supporting students' engagement [52, 53]. In addition to challenges experienced by all students during the COVID-19 pandemic, refugee students face unique struggles with integration and engagement due to pre-existing educational challenges, potential trauma exposure, and stressors related to the resettlement process [54, 55]. Marginalized communities like refugee students have limited opportunities to form trustful relationships that could support their educational needs [56].

In the light of study findings and previous research, exploring ways to expand refugee students' opportunities to form virtual, trustful, supportive relationships is an important area for future research. Interventions that promote family involvement and students' engagement in offline and online school settings among refugees must be tailored, taking into consideration their unique contextual factors discussed above. Interventions with families, caring adults, and students to increase youth engagement in school have been implemented, including the Family Check-Up model, which focuses both on parent participation in education and youth academic achievements, resulting in improved youth self-regulation and decreased depression [57, 58]. Models developed to address student engagement, such as the Check and Connect model of "identification, treatment and skill building among individuals and families," could be appropriate. This model includes core elements of relationship-building, ensuring active affiliation with the school, and encouraging a problem-solving approach to student wellbeing and achievement [58]; the core concerns voiced by service providers indicated that an active approach to identifying students at-risk of disengagement or dropping out of school will be needed in the transition back to in-person education. Such interventions could be adapted and piloted to contexts addressing MENA-background students' and families' needs in the context of COVID-19. In addition, educators and other school-affiliated staff reported high levels of stress due to the demands of teaching and providing other support services during school closures and concerns regarding the long-term impacts of the pandemic on their students' learning and wellbeing. Emerging research indicates that teachers have struggled to engage students and keep them motivated, which has impacted their own perceptions of self-efficacy and reinforced feelings of isolation, and that burnout is a major concern as the pandemic, and its attendant impacts on education, continues [59]. Special attention should be paid to teachers' and providers' training in supporting and communicating with students and their families in the virtual learning context.

These data should be interpreted in light of some limitations. Firstly, given difficulties recruiting students and caregivers to participate in SALaMA during school closures, this sample consists solely of service providers. Therefore, results can be interpreted as service providers' perceptions of the impact of the pandemic on student mental health, for example, whereas primary data collection with students and caregivers is key to validate these findings. Inclusion of perspectives from youth and caregivers may have indicated differences in experiences and relative importance of different factors for students. Secondly, the sites were selected purposively, based on prior relationships and ongoing data collection for SALaMA, and therefore results cannot be generalized to other contexts of service providers and refugee students during the pandemic. Future research could consider the challenges and strengths that were common across the three diverse study sites but must also consider the unique community needs and protective factors present for MENA students in their local context.

## Conclusion

Evidence concerning the immediate and long-term impact of the COVID-19 pandemic on immigrant and refugee students' well-being and learning outcomes is as yet not definitive, and

will emerge in the coming years. Initial research indicates that vulnerable groups, including immigrants and refugees, in the United States, experienced increased impacts, including socio-economic stressors, loss of critical language learning and mental health supports in educational settings, and higher levels of disease and death, during the pandemic. This study explored the perspective of service providers and describe and analyze their efforts to provide high-quality educational and mental health services for immigrant and refugee students from the MENA region in Harrisonburg, Detroit and Chicago. These data indicate that service providers perceived significant impacts of the pandemic on the mental health and well-being of immigrants and refugees from the MENA region, and changes in access to and utilization of key services. Despite their best efforts to provide for, connect with and support students, service providers described a context of reduced services, social isolation and high levels of stressors for students and their families. In the context of acute and long-term effects of the pandemic on education systems in the United States, these data should inform policy and programming to support immigrant and refugee students' wellbeing and learning.

## Author Contributions

**Conceptualization:** Sarah R. Meyer, Ilana Seff, Lindsay Stark.

**Data curation:** Sarah R. Meyer, Alli Gillespie, Hannah Brumbaum, Najat Qushua.

**Formal analysis:** Sarah R. Meyer, Ilana Seff, Alli Gillespie, Hannah Brumbaum, Najat Qushua.

**Funding acquisition:** Lindsay Stark.

**Investigation:** Sarah R. Meyer, Alli Gillespie, Hannah Brumbaum, Najat Qushua.

**Methodology:** Ilana Seff, Lindsay Stark.

**Supervision:** Ilana Seff, Lindsay Stark.

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
