## [Decision Letter · Decision Letter 0]

27 Jun 2022

PONE-D-22-05430

“We will need to build up the atmosphere of trust again”: Perceptions of COVID-19 impacts on resettled refugee adolescents and educational and support services in the United States

PLOS ONE

Dear Dr. Stark,

Thank you for submitting your manuscript to PLOS ONE. After careful consideration, we have decided that your manuscript does not meet our criteria for publication and must therefore be rejected.

The paper has not been prepared according to the journal’s guidelines and is not well structured. Further, the paper does not make a contribution to any new dimensions, concepts or knowledge in the discipline. Even the Introduction section fails to engage in bringing out the motivation/rationale to the research investigation.

The authors have not demonstrated on the reliability and validity of the instrument used for the purpose of the study. Overall the analysis presented is inadequate.

I am sorry that we cannot be more positive on this occasion, but hope that you appreciate the reasons for this decision.

Kind regards,

Prabhat Mittal, Ph.D.

Academic Editor

PLOS ONE

Reviewers' comments:

Reviewer's Responses to Questions

**Comments to the Author**

1. Is the manuscript technically sound, and do the data support the conclusions?

Reviewer #1: Partly

Reviewer #2: Yes

2. Has the statistical analysis been performed appropriately and rigorously? 

Reviewer #1: Yes

Reviewer #2: N/A

3. Have the authors made all data underlying the findings in their manuscript fully available?

Reviewer #1: Yes

Reviewer #2: No

4. Is the manuscript presented in an intelligible fashion and written in standard English?

Reviewer #1: Yes

Reviewer #2: Yes

5. Review Comments to the Author

Reviewer #1: To Author(s).

Date 28/04/2022

Reviewer comments

Dear /sir, I hope this message find you well. These are Constructive Comments forwarded for the Author(s).i will appreciate for your concern if you address the given comments, questions or suggestion on time.

The topic is very interesting which shows the impact COVID-19 on the refuge settlements that helps the issues to contribute for the tackling of the pandemic of COVID-19. In order to make your manuscript to the standards of scientific community in sharing knowledge and experiences of your work with other scientific or researchers the following points are forwarded. Overall Check for grammars/spellings error carefully. : re writes and make the title short beside this Make: COVID-19 impact on educational and support services for resettled adolescent refugee

Make keywords: less or = 5, use plos one guide line while writing the abstract…generally.

Avoid; the use of”we” ‘our’ …rather use the researcher /investigators/author[s].

Under the methods explain the benefit, disadvantage , and reasons for choosing content analysis in relation to your study.

we aimed: to describe the perspective of educators and other school-affiliated service providers on the impact of the COVID-19 pandemic on mental health and wellbeing of adolescent resettled refugees; to examine educators’ and other service providers’ views on the impact of the COVID-19 pandemic on access to and quality of education and support services for adolescent resettled refugees; and to explore adaptations and innovations in service delivery in the context of the COVID-19 pandemic

Please make the objective very concise, and brief, that means make the general objective broad rather than breaking in to pieces.

In writing Abstract state the elements /components to be included like introduction/background, objective, methods, results…conclusion.

Selects terms while writing’ Our findings’ better to use the current study finding , this study , the finding of this study….

Give space after every paragraph e.g. Pp: 13 and discussion and conclusion part need space over all.

please upload the revised Manuscript in the form of (DOC, DOCX), Regarding references I hope you did it that was nice. Put cation for accordingly, too in the text where to be cited

Old reference >>>recent one

Hsieh H-F, Shannon SE. Three Approaches to Qualitative Content Analysis. Qual Health Res. 2005 Nov 1;15(9):1277–88.

Sinclair MF, Christenson SL, Lehr CA, Anderson AR. Facilitating Student Engagement: Lessons Learned from Check & Connect Longitudinal Studies. Calif Sch Psychol. 2003;8:29–41.

Good luck

Thanks for your response

JA Ahmed

Reviewer #2: This is an interesting and well-written paper on an important topic. The author does a good job of describing the literature describing how the impacts of COVID-19 may have been different and/or more severe for refugee youth.

The methods proposed are reasonable under the circumstances; as the author notes, it would have been helpful to hear from students about how and why they found the situation challenging rather than relying only on the perceptions and beliefs of teachers and service providers. However, COVID made recruitment difficult and this information is still important.

The analyses are described quite well but I could not find the two tables that are referred to in the text. I thought perhaps that they may be a separate appendix, but there did not seem to be one for this paper.

The results focused mostly on the impact of remote learning, which is not surprising given that the educational system was the starting point for snowball sampling and thus would have central to the participants' perceptions of the impact of the COVID pandemic on youth. Schools are in fact a central institution in the well-being of youth and so this may have been the outcome regardless of who was interviewed, but the importance of family also emerged as a source of support, stress, and responsibility. I wonder if having interviewed the youth or their parents might have given us a different balance in terms of relative importance? The observation about how being forced to let students work independently was an asset for some students was interesting, as was the multiple observations about how many students don't participate or communicate in classrooms or on-line but then recommendations to provide open forums for students to communicate. These contradictions could perhaps be explored further.

I found the discussion a little difficult and the weakest part of the paper. It is comprised of very long paragraphs that seem to just list information rather than building arguments. I think that this section could have been stronger if it did not just summarize the findings but sought to bring a stronger analytic lens to the findings, reflecting more on implications for education in refugee youth well-being, conditions under which COVID had greater or lesser impact, unique challenges for refugees, etc. I actually think that some or even all of this is there but it doesn't come through as clearly as it could

6. PLOS authors have the option to publish the peer review history of their article (what does this mean?). If published, this will include your full peer review and any attached files.

Reviewer #1: **Yes: **JA Ahmed

Reviewer #2: No

- - - - -

---

## [Author Response · Author response to Decision Letter 0]

26 Jul 2022

Response to Reviewers: PONE-D-22-05430, ““We will need to build up the atmosphere of trust again”: Perceptions of COVID-19 impacts on resettled refugee adolescents and educational and support services in the United States” 

To the Editors, PLoS One: 

Thank you for the recognition of the contribution of our manuscript, ““We will need to build up the atmosphere of trust again”: Perceptions of COVID-19 impacts on resettled refugee adolescents and educational and support services in the United States.” In response to the reviewers’ comments, several changes have been made, as detailed below. We feel that the suggestions have greatly helped improve this manuscript and thank the reviewers for their feedback and comments. The reviewers’ comments are addressed point-by-point in turn below. 

Reviewer 1: 

1. Overall Check for grammars/spellings error carefully.

We have reviewed the manuscript carefully for any spelling or grammatical errors

2. Rewrite and make the title short beside this Make: COVID-19 impact on educational and support services for resettled adolescent refugee

Thank you for this comment. The authors feel that inclusion of the quote in the article title is useful and interesting for the reader. In addition, the author feel that it is important to include the phrase “perceptions of impacts” to clarify that this is a qualitative study of the perceptions of key informants, rather than a quantitative survey that assesses the impacts on specific outcomes. 

3. Make keywords: less or = 5, use plos one guide line while writing the abstract…generally.

We have reduced the number of keyword and revised the abstract 

4. Avoid; the use of”we” ‘our’ …rather use the researcher /investigators/author[s].

We have edited the manuscript throughout to address this comment

5. Under the methods explain the benefit, disadvantage , and reasons for choosing content analysis in relation to your study.

We have added the following to respond to this useful comment: 

“Directed content analysis was selected as the analytic method as it fit the objectives of the study; one of the benefits of this approach is that it can be used to expand on or further describe a phenomenon or set of phenomena about which there is already some theory or research, as is the case in this study (43). While directed content analysis may result in bias due to use of a predetermined theory or categories, in the case of the present analysis, the initial codes were drawn directly from data.”

6. Please make the objective very concise, and brief, that means make the general objective broad rather than breaking in to pieces.

We have revised the objective based on this feedback, and it now reads: 

“The objective of this study was to describe the perspective of educators and other school-affiliated service providers on the impact of the COVID-19 pandemic on mental health and wellbeing of adolescent resettled refugees and access to and quality of education and support services for adolescent resettled refugees.”

7. In writing Abstract state the elements /components to be included like introduction/background, objective, methods, results…conclusion.

We have added sub-headings to the Abstract 

8. Selects terms while writing’ Our findings’ better to use the current study finding , this study , the finding of this study….

We have edited the manuscript throughout to address this comment

9. Give space after every paragraph e.g. Pp: 13 and discussion and conclusion part need space over all.

We have added spaces after each paragraph.

10. Add citations: Hsieh H-F, Shannon SE. Three Approaches to Qualitative Content Analysis. Qual Health Res. 2005 Nov 1;15(9):1277–88.

Sinclair MF, Christenson SL, Lehr CA, Anderson AR. Facilitating Student Engagement: Lessons Learned from Check & Connect Longitudinal Studies. Calif Sch Psychol. 2003;8:29–41.

The Hsieh citation is already included (citation Number 43). 

The Sinclair et al. paper is a useful addition and we integrated this citation into the Discussion section. 

Reviewer 2: 

1. The methods proposed are reasonable under the circumstances; as the author notes, it would have been helpful to hear from students about how and why they found the situation challenging rather than relying only on the perceptions and beliefs of teachers and service providers. However, COVID made recruitment difficult and this information is still important.

Thank you – we had intended to conduct data collection with students, but COVID 19’s impact on schools made this impossible.

2. The analyses are described quite well but I could not find the two tables that are referred to in the text. I thought perhaps that they may be a separate appendix, but there did not seem to be one for this paper.

We have included the two tables now.

3. The results focused mostly on the impact of remote learning, which is not surprising given that the educational system was the starting point for snowball sampling and thus would have central to the participants' perceptions of the impact of the COVID pandemic on youth. Schools are in fact a central institution in the well-being of youth and so this may have been the outcome regardless of who was interviewed, but the importance of family also emerged as a source of support, stress, and responsibility. I wonder if having interviewed the youth or their parents might have given us a different balance in terms of relative importance? The observation about how being forced to let students work independently was an asset for some students was interesting, as was the multiple observations about how many students don't participate or communicate in classrooms or on-line but then recommendations to provide open forums for students to communicate. These contradictions could perhaps be explored further.

We agree that interviews with youth and parents may have indicated a different balance in terms of relative importance – we have added another sentence in the Limitations to indicate that we recognize this: 

“Inclusion of perspectives from youth and caregivers may have indicated differences in experiences and relative importance of different factors for students.” 

In addition, we have added the following in the Discussion section to highlight the contradiction noted: 

“Multiple contradictions emerged regarding online programming, participation and wellbeing; for some service providers, the pandemic spurred much-needed independence and self-sufficiency for MENA-background students, while for others, students’ nearly total lack of participation in online programming led to immense learning loss and social isolation.” 

4. I found the discussion a little difficult and the weakest part of the paper. It is comprised of very long paragraphs that seem to just list information rather than building arguments. I think that this section could have been stronger if it did not just summarize the findings but sought to bring a stronger analytic lens to the findings, reflecting more on implications for education in refugee youth well-being, conditions under which COVID had greater or lesser impact, unique challenges for refugees, etc. I actually think that some or even all of this is there but it doesn't come through as clearly as it could

Thank you -we have edited, streamlined and restructured the Discussion section, in line with these comments.

---

## [Decision Letter · Decision Letter 1]

16 Feb 2023

PONE-D-22-05430R1

“We will need to build up the atmosphere of trust again”: Perceptions of COVID-19 impacts on resettled refugee adolescents and educational and support services in the United States

PLOS ONE

Dear Dr. Lindsay,

Thank you for submitting your manuscript to PLOS ONE. After careful consideration, we feel that it has merit but does not fully meet PLOS ONE’s publication criteria as it currently stands. Therefore, we invite you to submit a revised version of the manuscript that addresses the points raised during the review process.

We look forward to receiving your revised manuscript.

Kind regards,

Nelsensius Klau Fauk, S.Fil., M., MHID, MSc, PhD

Academic Editor

PLOS ONE

Journal Requirements:

4. Please include your tables as part of your main manuscript and remove the individual files. Please note that supplementary tables (should remain/ be uploaded) as separate "supporting information" files.

Additional Editor Comments (if provided):

Authors have sufficiently address the comments of reviewers and improved the manuscript. However, there are a few additional minor comments that need to be addresses.

Reviewers' comments:

Reviewer's Responses to Questions

**Comments to the Author**

1. If the authors have adequately addressed your comments raised in a previous round of review and you feel that this manuscript is now acceptable for publication, you may indicate that here to bypass the “Comments to the Author” section, enter your conflict of interest statement in the “Confidential to Editor” section, and submit your "Accept" recommendation.

Reviewer #2: All comments have been addressed

Reviewer #3: (No Response)

Reviewer #4: All comments have been addressed

Reviewer #5: (No Response)

Reviewer #6: All comments have been addressed

Reviewer #7: All comments have been addressed

Reviewer #8: All comments have been addressed

2. Is the manuscript technically sound, and do the data support the conclusions?

Reviewer #2: Yes

Reviewer #3: Partly

Reviewer #4: Yes

Reviewer #5: (No Response)

Reviewer #6: Yes

Reviewer #7: Yes

Reviewer #8: Partly

3. Has the statistical analysis been performed appropriately and rigorously? 

Reviewer #2: N/A

Reviewer #3: N/A

Reviewer #4: Yes

Reviewer #5: (No Response)

Reviewer #6: Yes

Reviewer #7: Yes

Reviewer #8: No

4. Have the authors made all data underlying the findings in their manuscript fully available?

Reviewer #2: No

Reviewer #3: (No Response)

Reviewer #4: Yes

Reviewer #5: (No Response)

Reviewer #6: Yes

Reviewer #7: (No Response)

Reviewer #8: No

5. Is the manuscript presented in an intelligible fashion and written in standard English?

Reviewer #2: Yes

Reviewer #3: Yes

Reviewer #4: Yes

Reviewer #5: (No Response)

Reviewer #6: Yes

Reviewer #7: Yes

Reviewer #8: Yes

6. Review Comments to the Author

Reviewer #2: The authors have addressed my concerns. The discussion is much clearer now. I have two thoughts that do not need to be addressed but that I wonder about; the data were collected at different points in the pandemic and I wondered if the themes differed based on when the data were collected. However, I think that they were also collected in different locations and under different circumstances as well so this may not be an aswerable question.

The other observation that repents felt that some students found working from home less distracting and I wondered if, for some students, the school is not a safe space and so attending remotely may have been a safer option.

I don't need or expect a response to either of these points....just musing....a very interesting paper!

Reviewer #3: Definitely a good and interesting manuscript. It is important to note, however, that some major and minor concerns exist:

1. Trustworthiness of the data analysis should be demonstrated.

2. In the data analysis: As the directed content analysis was used, the pre-existing theory should be described up front clearly.

3. In the discussion section: the authors should state if their findings offer supporting or non-supporting evidence for the pre-existing theory. In addition, any newly identified categories should be discussed in terms of whether they contradict or enrich the original theory.

4. The authors aim to demonstrate the impact of the COVID-19 pandemic on the mental health and wellbeing of adolescent resettled refugees; however, the data does not fully support this conclusion.

5. The data collection process was multiphase and part of another study [SALaMA]. For the readers to understand how the process evolved, it would be better if a flow chart was used to display the research journey clearly and concisely.

6. The title is misleading and does not accurately say what the study is about. It is better to use a more direct and clear title.

Regards

Reviewer #4: 1-Before analysis the data by using the Dedoose qualitative software, the authors should try-out in other group that is similar the sample group in this study. That garantee the data collection of this study is suitabale for using the Dedoose qualitative software.

2-In the result part of this study the authors might summary the relationship between variables of this study.

Reviewer #5: Overall, this article is important, interesting, and useful. However, it should be improved a bit as follows.

The Abstract should be written in one paragraph, not divided into sub-headings.

Research articles from other parts of the world may also be included in the discussion. And these research articles should be considered for reference.

 Urairak, B. (2022). Effect of the COVID-19 Pandemic Related Mental Health on State Anxiety in Thailand. Asian Administration and Management Review, 5( 1), 1-6.

 Mustari, N., Herman, Aris, M., Mawardi, A., & Chaminra, T. (2021). The Effect of Online Learning Policy in the Era of Covid-19 on Students’ Quality. Asian Political Science Review, 5(2), 1-8.

Reviewer #6: The manuscript title: “We will need to build up the atmosphere of trust again”: Perceptions of COVID-19 impacts on resettled refugee adolescents and educational and support services in the United States., Overall, this manuscript is quite well written., It appropriate to publish to journal. Please recheck all references, table and figure again.

Reviewer #7: “We will need to build up the atmosphere of trust again”: Perceptions of COVID-19 impacts on resettled refugee adolescents and educational and support services in the United States. Overall concise, interesting and well written. Please recheck all references styles.

Reviewer #8: The study is an opinion article about perceptions of interviewees, rather than a data driven

quantitative study to assesses the outcomes. The title is an opinion expression rather than indicator of study’s key features. Authors should consider revising the title itself. The study has not factored in the dichotomy created by inputs from heterogenous populations of youth and caregivers, which may confound any conclusions driven.

7. PLOS authors have the option to publish the peer review history of their article (what does this mean?). If published, this will include your full peer review and any attached files.

Reviewer #2: No

Reviewer #3: No

Reviewer #4: No

Reviewer #5: **Yes: **

Reviewer #6: No

Reviewer #7: No

Reviewer #8: No

---

## [Author Response · Author response to Decision Letter 1]

6 Mar 2023

Please see attached reviewer comment table.

---

## [Editor Report · Decision Letter 2]

13 Mar 2023

“We will need to build up the atmosphere of trust again”: Service providers' perceptions of experiences of COVID-19 amongst resettled refugee adolescents

PONE-D-22-05430R2

Dear Dr. Stark,

We’re pleased to inform you that your manuscript has been judged scientifically suitable for publication and will be formally accepted for publication once it meets all outstanding technical requirements.

Kind regards,

Nelsensius Klau Fauk, S.Fil., M., MHID, MSc, PhD

Academic Editor

PLOS ONE
---

## [Editor Report · Acceptance letter]

17 Mar 2023

PONE-D-22-05430R2 

“We will need to build up the atmosphere of trust again”: Service providers’ perceptions of experiences of COVID-19 amongst resettled refugee adolescents 

Dear Dr. Stark:

I'm pleased to inform you that your manuscript has been deemed suitable for publication in PLOS ONE. Congratulations! Your manuscript is now with our production department. 

Kind regards, 

on behalf of

Dr. Nelsensius Klau Fauk 

Academic Editor

PLOS ONE